# Formalin-free tissue embedding is less hazardous and results in better DNA quality

**Maarten Niemantsverdriet, Ageeth Knol, Jose van der Starre-Gaal, Agnes Marije Hoogland** *

Isala Pathology, Zwolle, The Netherlands

* a.m.hoogland@isala.nl

**Data Availability Statement:** All data is submitted in supplementary data.

**Funding:** The author(s) received no specific funding for this work.

## Abstract

Disease diagnosis, prognosis, and therapy choice progressively rely on good-quality Deoxyribonucleic acid (DNA) for molecular analysis. As tissue processing is routinely performed worldwide with ancient techniques using toxic and DNA-damaging formaldehyde, the quality of DNA isolated from embedded tissues used for diagnostics is poor. We used a novel formalin-free tissue embedding method to process tissues and show that, after 5 years, DNA quality is superior to formalin fixation.

## Introduction

As the pathology field is ever more dependent on molecular data for diagnosis, prognosis, and therapy choice; good quality tissue DNA for molecular analysis is increasingly more important [1]. Processing patient tissue samples by pathology laboratories is routinely done by formalin fixation and paraffin embedding (FFPE) to preserve tissue morphology and enable immuno-histochemical analysis [2, 3]. The basis of formalin fixation was established already in 1868 with the use of formaldehyde for gross anatomy and microscopic techniques [4]. Formaldehyde, the key component of formalin, is very hazardous and can cause acute oral toxicity, acute dermal toxicity, acute inhalation toxicity, skin corrosion/irritation, serious eye damage/eye irritation, skin sensitization, germ cell mutagenicity, carcinogenicity, and specific target organ toxicity [5]. Additionally, genomic analysis of DNA extracted from FFPE blocks, especially from blocks over three years old, is problematic, as formalin fixation negatively impacts DNA quality and quantity compared to fresh frozen (FF) material [6]. Downstream genomic applications are hampered by issues caused by formalin, such as 1) DNA fragmentation, leading to diminished amounts of amplifiable template for Polymerase Chain Reaction (PCR) amplification, 2) stalling of polymerases and inhibiting denaturation, thereby reducing amplification efficacy and 3) causing non-reproducible C>T/G>A sequencing artifacts that can result in the misinterpretation of mutation analysis by calling false-positive mutations or filtering away true clinically relevant mutations hidden in the background of artifacts [6–8]. As such, the ancient technique of formalin-fixation does not meet today's safety and DNA quality requirements. The solution to this would require a drastic (global) change in the way tissues are routinely processed. To this end, we used a novel tissue processing method [9], removing

**Competing interests:** I have read the journal"s policy ans the autors of this manuscript have the following competing interest: M.H. TISPA patent pending, M.N., J.G., and A.K. declare no competing interests.

formaldehyde from the process and using supercritical Carbon dioxide ($CO_2$) to create non-formalin processed paraffin-embedded tissues (NFPE) and we analyzed the DNA 5 years later.

## Methods and materials

### Tissue processing

Formalin-fixed paraffin-embedded (FFPE) tissue processing was performed using standard methods. Non-formalin-fixed paraffin-embedded (NFPE) tissue processing was performed using supercritical carbon dioxide ($CO_2$), basically, as the alternative embedding technique described for formalin-fixed tissues by Bluel et al. [9]. In our study, fresh, unfixed tissues arriving from the operation room were assessed and processed by a pathologist and split. Three tissues were accidentally pre-incubated in 35% Ethanol (EtOH) for 2 hours before splitting (this was a mistake by an intern, we kept the tissues for research purposes only). Of each tissue, one sample was used for FFPE gold standard and the other sample for the fresh non-formalin processed paraffin-embedded (NFPE) route (Fig 1). The schematic processes with timelines for both NFPE and FFPE, the machines used, and representative resulting paraffin blocks are presented in Fig 1.

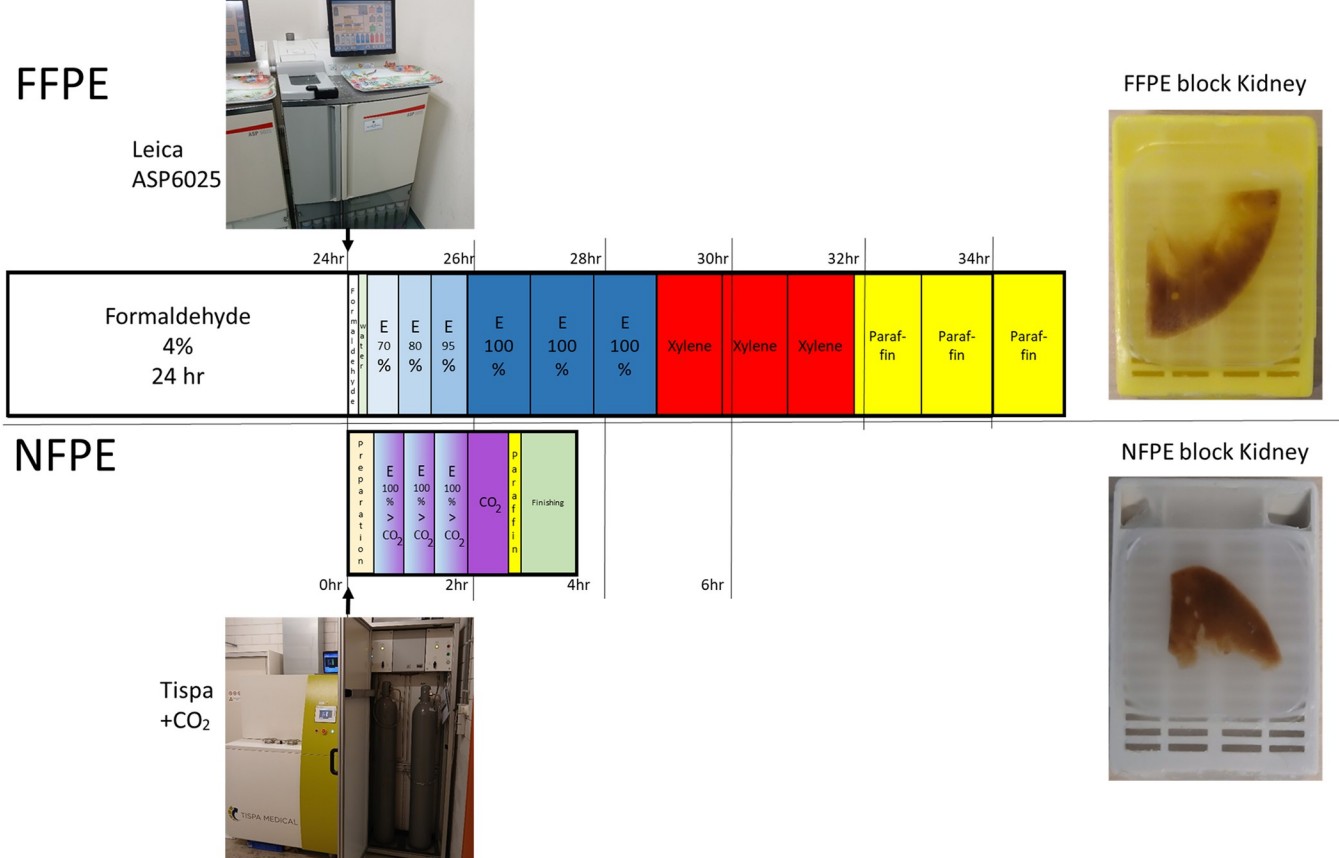

**Fig 1. Comparison and schematic timelines of FFPE and NFPE tissue processing.** For FFPE, tissues were pre-incubated in formaldehyde 4% (formalin) for 24 hrs and processed as paraffin blocks in a Leica ASP6025 using standard protocol (picture). For NPFE samples, tissue was loaded directly (unfixed) into the Tispa machine equipped with $CO_2$ tanks (picture) and processed as paraffin blocks. FFPE and NFPE processes are described in detail below including times, temperatures, and pressure (NFPE) for each step. E = Ethanol. Resulting representative blocks of split kidney tissue, one part processed as FFPE (above, yellow cassette) and one part as NFPE (below, grey cassette) are shown. Blocks shown are over 5 years old.

For the FFPE route, the sample was incubated in formalin (4% formaldehyde) and processed with the Leica ASP6025 using the following protocol: Formaldehyde 4% (5min., 37˚C), Water rinse (2min., 37˚C), Ethanol 70% (30 min., 35˚C), Ethanol 80% (30 min., 35˚C), Ethanol 95% (30 min., 35˚C), 3 x Ethanol 99% (60 min., 35˚C), 3 x Xylene (60 min., 40˚C), paraffin (60 min., 65˚C), 2x paraffin (90 min., 65˚C), all steps at normal pressure.

For the NFPE route, fresh, unfixed tissue was loaded in the vessels of the Tispa tissue processor (Tispa Medical) and processed using the "Large sample" protocol. First, the system was prepared by pumping paraffin in buffer vessels of the machine (4 min., 60˚C, 0 bar), pumping ethanol in buffer vessels of the machine (2 min., 60˚C, 0 bar), bringing the (tissue)vessels to a pressure of 150 bar (3 min, 54˚C, 150 bar) and keeping at a pressure of 150 bar (20 min., 55˚C, 150 bar). Next, 3 cycles of EtOH/$CO_2$ gradient steps were applied (25 min each): system was prepared to pump ethanol to the (tissue) vessel (3 min., 56˚C, 150 bar), ethanol 99–100% was pumped in the vessel holding the tissue (3 min., 57˚C, 150 bar), after which $CO_2$ was pumped in, mixing ethanol and $CO_2$ (3 min., 58˚C, 142 bar), dewatering the tissue by holding at pressure (15 min., 58˚C, 142 bar) and pressure was raised again to 150 bar (1 min. 59˚C, 150 bar). After the 3 cycles of EtOH/$CO_2$ gradients, fresh $CO_2$ was pumped in to rinse the tissues from ethanol (30 min, temp gradient 60–64˚C, 150 bar). Next, $CO_2$ was pumped in at lower pressure (4min., 64˚C, 135 bar). Paraffin was pumped in (5 min., 64˚C, 135 bar) and held at pressure (5 min. 64˚C, 135 bar). After the paraffin step, the process was finalized by gradually depressurizing the vessels from 135 to 0 bar (18 min., 64˚C, 135–0 bar) and a vacuum step (30 min, 64˚C, -1 bar).

For both FFPE and NFPE, after embedding at the paraffin embedding station, blocks were placed on a cold plate to solidify. The resulting blocks from both NFPE and FFPE were stored at room temperature in the dark and left alone for 5 years. NFPE and FFPE blocks were stored under identical conditions. From both samples (FFPE and NFPE), 12 3μm slides were cut and the first (before HE) and last (after HE) were stained with Hematoxylin and Eosin (HE) to assess the amount of vital tissue for cell surface measurements.

## Molecular analysis

DNA was isolated from the 10 slides between the before and after HE using the Maxwell® CSC Instrument (Promega), using Instructions for use of the Maxwell® CSC Instrument when running IVD Mode, Model Number AS6000. DNA concentration was measured using a Qubit 4.0. To calculate the tissue surface corrected-DNA yield, sections stained with HE were scanned. The tissue surface was measured using PDSP (Philips) software. The DNA concentration was divided by the tissue surface for the NFPE and the FFPE samples. The DNA/surface of NFPE was divided by the DNA/surface of FFPE.

Ladder PCR was performed on 5 ng DNA for 35 cycles, using 4 primer sets producing bands of 100bp (TBXAS1), 200bp (RAG1), 300bp (PLZF), and 400bp (AF4), respectively, as described by van Dongen et al. [10]. 5μl PCR products were visualized on 2% agarose gels. 267 amplicon Next-Generation-Sequencing (NGS) panel libraries were made using the Oncomine Focus DNA Ampliseq 16 kit (Thermofisher) and Ion Chef System (Thermofisher) according to the manufacturer's instructions. The library yield was quantified using the Ion Library Taqman Quantitation kit (Thermofisher) (library qPCR) according to the manufacturer's instructions. NGS was performed on an Ion GeneStudio S5 plus (Thermofisher) according to the manufacturer's instructions. Mean read length and read length histogram analysis were performed using Torrent Suite 5.12.1 software (Thermofisher). NGS was analyzed using Ion Reporter 5.12.3.0 software (Thermofisher). To analyze fixation artifacts, the "no filter" filter

chain was used. Formalin fixation artifacts were defined as C>T/G>A mutations in the range between 1 and 10%, as described by Wong et al. [6].

### Cost, processing time, and hands-on time

The Leica ASP6025 machine used for conventional embedding and the Tispa machine used for formalin-free embedding were in the same price range (exact prizes depend on purchasing year, contracts with manufacturer, discounts, etcetera). The total material costs of NFPE (Tispa) processing are approximately 20–30% less than conventional FFPE processing, mainly because no formalin and xylene were used. Since formalin and xylene are toxic, they have to be disposed of at a cost, which is not needed for NFPE. A great benefit of the NFPE method is the processing time, which is significantly shorter than that of FFPE. The total time required was less than 4 hours for NFPE whereas FFPE required over 35 hours (including a 24-hour formalin fixation step). The hands-on (technician) time is an estimated 5 times lower for NFPE processing since the process is much shorter and there are fewer fluids to be replenished.

### Guidelines and regulations

All methods were carried out in accordance with relevant guidelines and regulations. The Daily Board of the Medical Ethics Committee Isala Zwolle (The Netherlands), has reviewed the above-mentioned research proposal with METC file nr. 180107. As a result of this review, the Committee informs you that the rules laid down in the Medical Research Involving Human Subjects Act (also known by its Dutch abbreviation WMO), do not apply to this research proposal. The need for informed consent documentation was waived and all experimental protocols were approved as part of the ethics committee approval.

## Results

Ten tissues were split and one part was processed as FFPE blocks and one part as NFPE blocks (Fig 1). Blocks were stored in the conventional archive for 5 years, after which DNA was extracted and DNA parameters were tested (Figs 2 and 3 and Tables 1–4).

### DNA yield

The tissue surface corrected DNA yield was higher in 9 of 10 tissues for NFPE samples and in 1 tissue (liver) the relative yield was similar (Table 1). The DNA yield per tissue surface depends on tissue type and therefore the one-on-one comparison of NFPE and FFPE within

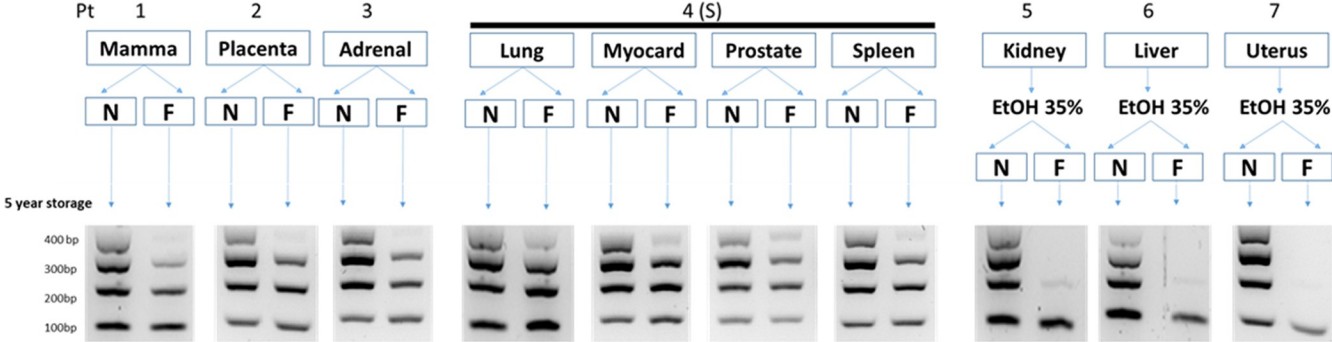

**Fig 2. Setup of the 5-year study: 10 tissues, derived from 7 patients were split and one part was processed as NFPE and the other as FFPE.** Three tissues were pre-incubated with EtOH 35% before splitting. After 5 years in storage, DNA was isolated and ladder PCR was performed.

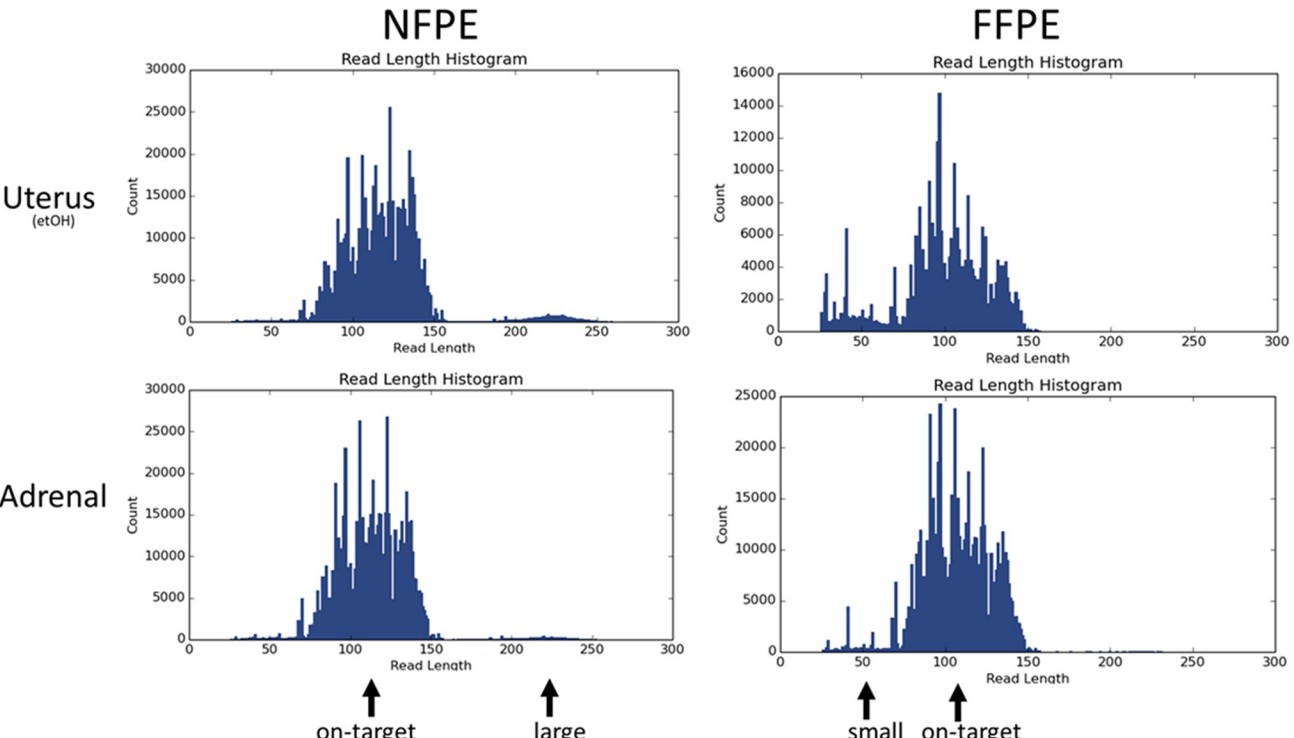

**Fig 3. Read length histograms of uterus (EtOH treated) and adrenal tissues, processed as either NFPE or FFPE.** On-target is the DNA size of library expected for this NGS panel. Small is smaller than the expected DNA size and large is larger than the expected DNA size.

the same tissue is more relevant than the comparison between tissue types. In this study the DNA yield per tissue surface ratio ranged from 1 to 58 fold, with an average of 8,3 fold more DNA isolated per tissue surface from NFPE than for FFPE.

## DNA fragmentation

Ladder PCR showed that the DNA of the NFPE material was less fragmented compared to the FFPE material for all 10 tissues, as shown by the largest band (400bp), which was clearly visible in all NFPE samples whereas it was not present or at much lower density in the FFPE samples (Fig 2). Three tissues (kidney, liver, and uterus) that were pre-incubated with EtOH 35% before splitting into a NFPE and FFPE sample, also did not show the 300bp and a faint 200bp band in FFPE samples whereas the 35% EtOH incubated NFPE samples showed clear 200bp, 300bp and 400bp bands in addition to the 100bp band seen in all samples (Fig 2).

## NGS library qPCR

Since the NGS library construction process requires multiple enzymatic reactions using polymerases and denaturation steps [11], steps inhibited by the formalin used in the FFPE process [12], we prepared NGS libraries of the DNA isolated from the NFPE and FFPE samples. After NGS library preparation, the libraries for each sample were quantified using qPCR, and the relative amount of library was calculated by dividing the NFPE value by the FFPE value. Library DNA of all 10 NFPE samples was amplified more efficiently compared to the corresponding FFPE samples. Three of these samples (placenta, kidney, and uterus) had exceptionally better NFPE libraries (Table 2). The fold increase in library qPCR ranged between 1,2 and 59 fold, with an average 14,7 fold higher amplification for NFPE samples.

**Table 1. DNA yield ratio per tissue surface of 5-year-old tissues processed as NFPE and FFPE.** The NFPE/FFPE ratio is shown. Patient 4 was a deceased (S) patient where 4 tissues were harvested. At a ratio > 1.0 the best result is for NFPE (< 1.0 best result for FFPE, not applicable).

| Tissue | Sample | Patient | Pretreatment | NFPE/FFPE | Best result |
|--------|--------|---------|--------------|-----------|-------------|
| | | | | ratio DNA yield | |
| Mamma | 1 | 1 | None | 6,0 | NFPE |
| Placenta | 2 | 2 | None | 2,1 | NFPE |
| Adrenal | 3 | 3 | None | 2,5 | NFPE |
| Lung | 4 | 4 (S) | None | 1,5 | NFPE |
| Myocard | 5 | 4 (S) | None | 1,3 | NFPE |
| Prostate | 6 | 4 (S) | None | 3,8 | NFPE |
| Spleen | 7 | 4 (S) | None | 2,2 | NFPE |
| Kidney | 8 | 5 | 35% EtOH | 4,4 | NFPE |
| Liver | 9 | 6 | 35% EtOH | 1,0 | similar |
| Uterus | 10 | 7 | 35% EtOH | 58 | NFPE |
| | | | average | 8,3 | |

## Mean read length

After library quantification, NGS using a small commercial hotspot panel was performed, a clear difference in mean read length (MRL) was noticed between NFPE and FFPE material. Of all 10 tissues, the NFPE sample had a higher MRL than the FFPE counterpart (Table 3). The average read length of NFPE was 116 bp whereas the average mean read length for FFPE samples was lower with 107 bp.

The difference in mean read length was visualized in the NGS histograms where FFPE samples showed more small library fragments and NFPE samples showed more large library fragments in addition to the on-target length fragments (two most illustrative tissues in Fig 3, the others in S1 Data).

## Sequencing artifacts

To assess if any difference in sequencing artifacts could be detected between NFPE and FFPE processing of the tissues, we analyzed unfiltered NGS data for the presence of C>T/G>A changes in the 1–10% Variant-allele-frequency (VAF) range [6]. In the first seven tissues, we

**Table 2. NGS library qPCR of 10 different tissues processed as NFPE and FFPE.** Library prep was performed using a small hotspot NGS panel generated with normalized DNA concentrations. MRL is mean read length. The NFPE/FFPE ratio is shown. Ratio > 1,0: best result is NFPE (Ratio < 1,0 FFPE, not applicable).

| Tissue | Sample | Patient | Pretreatment | NFPE/FFPE | Best result |
|--------|--------|---------|--------------|-----------|-------------|
| | | | | ratio qPCR | |
| Mamma | 1 | 1 | None | 3,0 | NFPE |
| Placenta | 2 | 2 | None | 31 | NFPE |
| Adrenal | 3 | 3 | None | 4,5 | NFPE |
| Lung | 4 | 4 (S) | None | 1,8 | NFPE |
| Myocard | 5 | 4 (S) | None | 1,8 | NFPE |
| Prostate | 6 | 4 (S) | None | 1,2 | NFPE |
| Spleen | 7 | 4 (S) | None | 1,5 | NFPE |
| Kidney | 8 | 5 | 35% EtOH | 59 | NFPE |
| Liver | 9 | 6 | 35% EtOH | 4,6 | NFPE |
| Uterus | 10 | 7 | 35% EtOH | 38 | NFPE |
| | | | average | 14,7 | |

**Table 3. NGS mean read length of 10 different tissues processed as NFPE and FFPE.** Best result is considered the lowest mean read length of the same tissue.

| Tissue | Sample | Patient | Pretreatment | N/F | MRL | Best result |
|--------|--------|---------|--------------|------|-----|-------------|
| Mamma | 1 | 1 | None | NFPE | 119 | NFPE |
|  |  |  |  | FFPE | 111 |  |
| Placenta | 2 | 2 | None | NFPE | 116 | NFPE |
|  |  |  |  | FFPE | 112 |  |
| Adrenal | 3 | 3 | None | NFPE | 113 | NFPE |
|  |  |  |  | FFPE | 106 |  |
| Lung | 4 | 4 (S) | None | NFPE | 117 | NFPE |
|  |  |  |  | FFPE | 112 |  |
| Myocard | 5 | 4 (S) | None | NFPE | 117 | NFPE |
|  |  |  |  | FFPE | 112 |  |
| Prostate | 6 | 4 (S) | None | NFPE | 116 | NFPE |
|  |  |  |  | FFPE | 111 |  |
| Spleen | 7 | 4 (S) | None | NFPE | 118 | NFPE |
|  |  |  |  | FFPE | 110 |  |
| Kidney | 8 | 5 | 35% EtOH | NFPE | 116 | NFPE |
|  |  |  |  | FFPE | 100 |  |
| Liver | 9 | 6 | 35% EtOH | NFPE | 112 | NFPE |
|  |  |  |  | FFPE | 95 |  |
| Uterus | 10 | 7 | 35% EtOH | NFPE | 116 | NFPE |
|  |  |  |  | FFPE | 97 |  |
|  |  |  | Average | NFPE | 116 |  |
|  |  |  |  | FFPE | 107 |  |

**Table 4. Sequencing artifacts in 10 different tissues processed as NFPE and FFPE.** C>T/G>A in the 1–10% VAF range of unfiltered NGS data are considered fixation artifacts. The best result is considered the lowest amount of artifacts of the same tissue.

| Tissue | Sample | Patient | Pretreatment | N/F | C>T/G>A | Best result |
|--------|--------|---------|--------------|------|---------|-------------|
| Mamma | 1 | 1 | None | NFPE | 0 | N/A |
|  |  |  |  | FFPE | 0 |  |
| Placenta | 2 | 2 | None | NFPE | 0 | N/A |
|  |  |  |  | FFPE | **1** |  |
| Adrenal | 3 | 3 | None | NFPE | 0 | N/A |
|  |  |  |  | FFPE | 0 |  |
| Lung | 4 | 4 (S) | None | NFPE | 0 | N/A |
|  |  |  |  | FFPE | 0 |  |
| Myocard | 5 | 4 (S) | None | NFPE | 0 | N/A |
|  |  |  |  | FFPE | 0 |  |
| Prostate | 6 | 4 (S) | None | NFPE | 0 | N/A |
|  |  |  |  | FFPE | 0 |  |
| Spleen | 7 | 4 (S) | None | NFPE | 0 | N/A |
|  |  |  |  | FFPE | 0 |  |
| Kidney | 8 | 5 | 35% EtOH | NFPE | 0 | NFPE |
|  |  |  |  | FFPE | **155** |  |
| Liver | 9 | 6 | 35% EtOH | NFPE | 0 | NFPE |
|  |  |  |  | FFPE | **135** |  |
| Uterus | 10 | 7 | 35% EtOH | NFPE | 0 | NFPE |
|  |  |  |  | FFPE | **201** |  |

did not observe serious C>T/G>A fixation artifacts in both the NFPE and the FFPE samples (just 1 in FFPE, placenta). However, in the three EtOH 35% pre-incubated tissues, the difference between NFPE and FFPE was remarkable, all three FFPE samples showed many C>T/G>A artifacts whereas all three matching NFPE samples lacked these artifacts completely (Table 4, details in S1 Data).

## Discussion

Formaldehyde (formalin) fixation is a standard procedure and routinely performed in pathology laboratories over the world. It has been demonstrated that DNA is damaged by formalin fixation, causing fragmentation, less efficient amplification by polymerases, non-reproducible sequencing artifacts and that those factors can be formed at varying proportions among different laboratories [6, 13]. We used a novel technique, non-formalin processed paraffin-embedded tissues (NFPE), to process tissues without formalin fixation and compared the DNA with DNA of the same tissues processed using the conventional FFPE method. We split ten different tissues, processed one part with and one part without formaldehyde, isolated DNA after 5 years of storage and performed basic DNA analysis (Figs 1 and 2).

The relative DNA yield of NFPE processed tissues was higher in 9 of 10 specimens (Table 1). The ladder PCR was better for NFPE processed tissues in all 10 out of 10 specimens (Fig 2). The mean read length was better for NFPE-processed tissues in all 10 out of 10 specimens (Table 2). The library qPCR was better for NFPE-processed tissues in all 10 out of 10 specimens (Table 3). Despite the fact that only ten tissues were used in this study, these 39/40 datapoints all suggest that better quality DNA is isolated from NFPE than from FFPE. Furthermore, although tested in a very artificial situation, the absence of fixation artifacts in NFPE-processed tissues incubated with 35% EtOH compared to many in FFPE samples incubated with 35% EtOH (Table 4) indicates that NFPE samples are less likely to acquire sequencing artifacts. In literature, high numbers of formalin-induced C>T/G>A sequence artifacts have been reported in some studies [6, 14, 15], whereas others report hardly any artifacts in FFPE [6, 16, 17]. In our own experience, for most FFPE blocks formalin artifacts are not an issue but occasionally a "bad block" appears that has a very high amount of artifacts. This implies that pre-analytical factors play a role in the predisposition to obtain formalin-induced damage. It is still largely unknown which factors are most important for the different types of formalin-associated types of DNA damage, these may include: tissue type, operator experience, time from tissue extraction to processing, tissue temperature, used preservatives, and processing protocol [6, 13, 18]. In the EtOH-treated tissues, the presence of many C>T/G>A variants in FFPE, in sharp contrast to no artifacts at all in NFPE (Table 4), indicates that these artifacts are not formed in NFPE tissue, not even when artificially predisposed to acquire C>T/G>A changes. The observation that qPCR is more efficient for NFPE DNA compared to matching FFPE (Table 2) may also indicate that stalling of polymerases and inhibiting denaturation is reduced in NFPE samples. However, even though DNA concentrations were normalized before libraries were made, it cannot be completely excluded based on our results that the difference is primarily caused by the degradation of DNA. Together, the data presented here show, using basic tests, that DNA quality is better in 5-year-old NFPE-processed tissue samples than in FFPE samples. To fully understand the impact of NFPE processing on molecular pathology, both DNA and RNA need to be analyzed in more detail in more specimens. Simultaneous quantification of DNA and RNA targets of different lengths and consequent calculation of the degradation index would allow to better characterize the impact of DNA and RNA degradation for downstream applications [19]. More elaborate research is also needed to fully understand to what extent (if any) the enhanced qPCR quantification and amplification

of the NFPE libraries can be attributed to reduced stalling of polymerases and reduced inhibition denaturation caused by formalin fixation in FFPE samples.

## Conclusion

Our results show that formalin-free processing of tissues (NFPE) is possible and shows that NFPE processing does not damage the DNA like FFPE processing. This suggests processing tissue as NFPE blocks may have the potential to drastically decrease the use of toxic formaldehyde in pathology labs and simultaneously improve DNA quality for molecular pathology. This study is a gateway to more elaborate DNA and RNA analysis using modern methods to fully understand the potential positive impact of NFPE processing on molecular pathology.

## Supporting information

**S1 Raw images.**
(PDF)

**S1 Data.**
(PDF)

## Acknowledgments

We thank Naomi van der Horst, Natalie Methorst, Inge Bruins, Savanna Hagen, and Marianne Roodbergen for technical assistance.

## Author Contributions

**Conceptualization:** Agnes Marije Hoogland.

**Data curation:** Ageeth Knol.

**Formal analysis:** Maarten Niemantsverdriet.

**Funding acquisition:** Jose van der Starre-Gaal, Agnes Marije Hoogland.

**Investigation:** Agnes Marije Hoogland.

**Methodology:** Maarten Niemantsverdriet, Agnes Marije Hoogland.

**Supervision:** Jose van der Starre-Gaal, Agnes Marije Hoogland.

**Writing – original draft:** Maarten Niemantsverdriet.

**Writing – review & editing:** Jose van der Starre-Gaal, Agnes Marije Hoogland.

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
