## [Decision Letter · Decision Letter 0]

23 Sep 2024

PONE-D-24-30519Formalin-free tissue embedding is less hazardous and results in better DNA qualityPLOS ONE

Dear Dr. Hoogland,

Thank you for submitting your manuscript to PLOS ONE. After careful consideration, we feel that it has merit but does not fully meet PLOS ONE’s publication criteria as it currently stands. Therefore, we invite you to submit a revised version of the manuscript that addresses the points raised during the review process.

**ACADEMIC EDITORshou/>==============================**

**A rebuttal letter that responds to each point raised by the academic editor and reviewer(s). You should upload this letter as a separate file labeled 'Response to Reviewers'.**

**A marked-up copy of your manuscript that highlights changes made to the original version. You should upload this as a separate file labeled 'Revised Manuscript with Track Changes'.**

**An unmarked version of your revised paper without tracked changes. You should upload this as a separate file labeled 'Manuscript'.**

**If applicable, we recommend that you deposit your laboratory protocols in protocols.io to enhance the reproducibility of your results. Protocols.io assigns your protocol its own identifier (DOI) so that it can be cited independently in the future. For instructions see: https://journals.plos.org/plosone/s/submission-guidelines#loc-laboratory-protocols. Additionally, PLOS ONE offers an option for publishing peer-reviewed Lab Protocol articles, which describe protocols hosted on protocols.io. Read more information on sharing protocols at https://plos.org/protocols?utm_medium=editorial-email&utm_source=authorletters&utm_campaign=protocols.

**We look forward to receiving your revised manuscript.**

**Kind regards,**

**Sherin Reda Rouby, PhD**

Academic Editor

**PLOS ONE**

**Additional Editor Comments (if provided)**:

Reviewers' comments:

**Reviewer's Responses to Questions**

**Comments to the Author**

1. Is the manuscript technically sound, and do the data support the conclusions?

**The manuscript must describe a technically sound piece of scientific research with data that supports the conclusions. Experiments must have been conducted rigorously, with appropriate controls, replication, and sample sizes. The conclusions must be drawn appropriately based on the data presented. **

**Reviewer #1: Partly**

**Reviewer #2: Yes**

**2. Has the statistical analysis been performed appropriately and rigorously? **

**Reviewer #1: No**

**Reviewer #2: Yes**

**3. Have the authors made all data underlying the findings in their manuscript fully available?**

**The PLOS Data policy requires authors to make all data underlying the findings described in their manuscript fully available without restriction, with rare exception (please refer to the Data Availability Statement in the manuscript PDF file). The data should be provided as part of the manuscript or its supporting information, or deposited to a public repository. For example, in addition to summary statistics, the data points behind means, medians and variance measures should be available. If there are restrictions on publicly sharing data—e.g. participant privacy or use of data from a third party—those must be specified.**

**Reviewer #1: No**

**Reviewer #2: Yes**

**4. Is the manuscript presented in an intelligible fashion and written in standard English?**

**PLOS ONE does not copyedit accepted manuscripts, so the language in submitted articles must be clear, correct, and unambiguous. Any typographical or grammatical errors should be corrected at revision, so please note any specific errors here.**

**Reviewer #1: Yes**

**Reviewer #2: Yes**

**5. Review Comments to the Author**

**Please use the space provided to explain your answers to the questions above. You may also include additional comments for the author, including concerns about dual publication, research ethics, or publication ethics. (Please upload your review as an attachment if it exceeds 20,000 characters)**

**Reviewer #1: This manuscript highlights the advantages of the novel formalin-free tissue embedding method over the routine formalin fixation method to maintain DNA quality in stored human tissues for further diagnostic purposes. However, the study in the current form is not suitable for publication in PLOS ONE**.

Major concerns:

I. Materials and methods, Tissue processing:

1. The procedures of tissue processing are not described in sufficient details that permits re-simulation of the study. The protocols for both paraffin and non-paraffin fixation should be mentioned in full details and documented with images that shows the tissue, the instruments, the steps of fixation and the final product after complete fixation, the methods of storage and the appearance of tissues after the 5 years duration.

2. A small sample size of only 10 tissue blocks limits the results of the study.

3. How and where the blocks were stored? and what were the conditions of storage for 5 years? Was there any regular check for the blocks during this period for any changes or deterioration?

Note that the methods and conditions of storage may act as confounders that affect the quality of DNA as well as the results of the study.

4. How was the tissue surface corrected-DNA concentration calculated?

II. Language: abbreviations used should be written for the first time encountered in the text (e.g. NGS, DNA, C>T/G>A, CO2, EtOH,....) the very frequent abbreviations make the manuscript difficult to read.

**III. Results: Data presentation is confusing, it is better to divide the result into subtitles, use more tables to demonstrate the data in numbers and apply statistical analysis whenever desirable**.

**IV. The discussion section is weak and needs to be elaborated upon with adding more related references.**

**Reviewer #2: 1. It would be prudent to address the cost differential between the 2 types of processing. What is the cost of each type of processor? What is the length of time on each processor? What is the technologist time required? Maybe NFPE has less toxic and has better DNA/molecular potential, but if it costs 5x more than FFPE, under-resourced areas will not be able to even entertain this new technique**.

2. Introduction - where "old blocks" is stated - what is the authors definition of old?

3. The methods section needs to be place above results, otherwise the paper is hard to follow.

4. A few more sentences are needed to explain the processing for NFPE. Another sentence or 2 on "standard FFPE" processing should also be included. i.e. FFPE processing involves placing specimen into formalin to fix the cells, dehydrating series of alcohols, clearing agent and final step of paraffin wax to solidify.

5. Discussion: list a few potential pre-analytical factors that play a role in the predisposition to obtain formalin-induced damage.

6. General grammatical errors: Extra periods, golden should be just gold.

**7. If 3 samples were accidently pretreated with etoh, why did the authors just not use them in this study? It would have been better if 10 were split and then as a second arm of the study, select another 10 specimens, preincubate with etoh and then split into nfpe and ffpe. This would have made the study and data stronger. It should be addressed why this decision to keep these specimens was made. (I also hope that the lab situation where issues like accidently putting tissues into wrong containers has been rectified as to not affect patient care.)**

**6. PLOS authors have the option to publish the peer review history of their article (what does this mean?). If published, this will include your full peer review and any attached files**.

**Reviewer #1: No**

**Reviewer #2: No**

****

**While revising your submission, please upload your figure files to the Preflight Analysis and Conversion Engine (PACE) digital diagnostic tool, https://pacev2.apexcovantage.com/. PACE helps ensure that figures meet PLOS requirements. To use PACE, you must first register as a user. Registration is free. Then, login and navigate to the UPLOAD tab, where you will find detailed instructions on how to use the tool. If you encounter any issues or have any questions when using PACE, please email PLOS at figures@plos.org. Please note that Supporting Information files do not need this step.**

---

## [Author Response · Author response to Decision Letter 0]

23 Oct 2024

Dear editor,

Thank you for allowing us to submit a revised version of our manuscript entitled; “Formalin-free tissue embedding is less hazardous and results in better DNA quality” by Maarten Niemantsverdriet, Ageeth Knol, Jose van der Starre-Gaal, and Marije Hoogland. We also thank the reviewers for their useful comments and suggestions. We have rewritten the manuscript and changed the vast majority of the points raised according to the reviewers’ suggestions. All points raised and the changes made in the revised manuscript are discussed below the reviewer’s comments. We uploaded a marked-up copy of our manuscript that highlights changes made to the original version (in yellow) as 'Revised Manuscript with Track Changes' and an unmarked version without tracked changes labeled 'Manuscript'. We believe that the manuscript has improved substantially and that it is now suitable for publication in Plos One. 

Sincerely,

Marije Hoogland

Reviewers' comments:

Reviewer's Responses to Questions

Comments to the Author

1. Is the manuscript technically sound, and do the data support the conclusions?

Reviewer #1: Partly

Reviewer #2: Yes

Response: In the revised manuscript we describe and discuss more clearly that our sample size is small but that all data point towards the conclusion that NFPE processing results in better DNA. 

2. Has the statistical analysis been performed appropriately and rigorously? 

Reviewer #1: No

Reviewer #2: Yes

Response: In the revised manuscript we included averages where possible.

3. Have the authors made all data underlying the findings in their manuscript fully available?

Reviewer #1: No

Reviewer #2: Yes

Response: No additional data are available. For NGS, the detailed information is in the supplementary data.

4. Is the manuscript presented in an intelligible fashion and written in standard English?

Reviewer #1: Yes

Reviewer #2: Yes

5. Review Comments to the Author

Reviewer #1: This manuscript highlights the advantages of the novel formalin-free tissue embedding method over the routine formalin fixation method to maintain DNA quality in stored human tissues for further diagnostic purposes. However, the study in the current form is not suitable for publication in PLOS ONE.

Major concerns:

I. Materials and methods, Tissue processing:

1. The procedures of tissue processing are not described in sufficient details that permits re-simulation of the study. The protocols for both paraffin and non-paraffin fixation should be mentioned in full details and documented with images that shows the tissue, the instruments, the steps of fixation and the final product after complete fixation, the methods of storage and the appearance of tissues after the 5 years duration.

Response: In the revised manuscript we added an extra figure with timelines for both the FFPE and the NFPE method, all steps schematically drawn, pictures of the machines used and examples of the final product after 5 years of storage. We also describe the steps for both the FFPE and the NFPE processing and the storage conditions in detail in the methods section. 

2. A small sample size of only 10 tissue blocks limits the results of the study.

Response: We agree that a larger sample size would have been much better and more research is needed as we state in the manuscript. However, the results presented in the paper are all pointing in the same direction. We looked at 5 DNA factors (ratio DNA yield, Ladder PCR, qPCR, MRL, and artifacts) in all 10 tissues. In total, 43 points could be evaluated (no sequencing artifacts were found for 7 specimens). Of these, 42 points were better for NFPE and one was equal. This shows that these data are very promising. We realize that further research is needed, and we mention that in the discussion, we now also mention that more specimens need to be investigated in future studies. 

3. How and where the blocks were stored? and what were the conditions of storage for 5 years? Was there any regular check for the blocks during this period for any changes or deterioration?

Note that the methods and conditions of storage may act as confounders that affect the quality of DNA as well as the results of the study.

Response: The blocks were stored at room temperature in the dark and they were left alone for 5 years. NFPE and FFPE blocks were stored under the same conditions. We now mention that in the methods section.

4. How was the tissue surface corrected-DNA concentration calculated?

Response: To calculate the tissue surface corrected-DNA concentration sections were stained with hematoxylin and eosin (HE) and photographed. The tissue surface was measured using PDPS (Philips) software. DNA was isolated from the 10 slides between the before and after HE and the DNA concentration was measured. The DNA concentration was divided by the tissue surface for both the NFPE and the FFPE samples. The DNA/surface of NFPE was divided by the DNA/surface of FFPE. For clarity, we now use the term DNA yield instead of concentration when more appropriate. The method is now written in full in the methods section. 

II. Language: abbreviations used should be written for the first time encountered in the text (e.g. NGS, DNA, C>T/G>A, CO2, EtOH,....) the very frequent abbreviations make the manuscript difficult to read.

Response: We agree with the reviewer that abbreviations need to be written in full the first time encountered in the text. In the revised manuscript we have written the abbreviations Next-Generation-Sequencing (NGS), Deoxyribonucleic acid (DNA), Polymerase Chain Reaction (PCR), Carbon dioxide (CO2), and Ethanol (EtOH) in full the first time in the manuscript. C>T/G>A is not written in full since it is not an abbreviation but the official way to represent formalin artifacts. We also agree that we often use abbreviations in the manuscript. This is mainly because we compare Deoxyribonucleic acid (DNA) of Formalin-fixed paraffin-embedded (FFPE) tissue and Non-formalin-fixed paraffin-embedded (NFPE) tissues using Next-Generation-Sequencing (NGS). These abbreviations are used very often, but they are too long to be written in full. That would not make the manuscript easier to read. 

III. Results: Data presentation is confusing, it is better to divide the result into subtitles, use more tables to demonstrate the data in numbers and apply statistical analysis whenever desirable.

Response: In the revised manuscript we split the table into 4 different tables, divided the results over subtitles, calculated averages when possible and discussed each result in more detail in a separate paragraph. We also split Figure 1 from the original manuscript into two figures (Figure 2 and 3).

IV. The discussion section is weak and needs to be elaborated upon with adding more related references.

Response: In the revised manuscript we have rewritten the discussion section and we used four additional references in the manuscript.

Reviewer #2: 1. It would be prudent to address the cost differential between the 2 types of processing. What is the cost of each type of processor? What is the length of time on each processor? What is the technologist time required? Maybe NFPE has less toxic and has better DNA/molecular potential, but if it costs 5x more than FFPE, under-resourced areas will not be able to even entertain this new technique.

Response: In the revised manuscript we estimated the cost, process time and hands-on time and described that in a new paragraph in the methods section. The novel method is cheaper and faster.

2. Introduction - where "old blocks" is stated - what is the authors definition of old?

Response: The older blocks get the more chance of problems with DNA quality. As this is usually only a problem in blocks older than 3 years, we now mention 3 years in the revised manuscript.

3. The methods section needs to be place above results, otherwise the paper is hard to follow.

Response: In the revised manuscript we placed the methods section above the results. 

4. A few more sentences are needed to explain the processing for NFPE. Another sentence or 2 on "standard FFPE" processing should also be included. i.e. FFPE processing involves placing specimen into formalin to fix the cells, dehydrating series of alcohols, clearing agent and final step of paraffin wax to solidify.

Response: In the revised manuscript we added an additional figure with timelines for both the FFPE and the NFPE method with all steps schematically drawn, pictures of the machines used, and examples of the final product after 5 years of storage. We also describe the steps for both the FFPE and the NFPE processes and the storage conditions in detail in the methods section. 

5. Discussion: list a few potential pre-analytical factors that play a role in the predisposition to obtain formalin-induced damage.

Response: In the revised manuscript we listed a few potential pre-analytical factors in the discussion.

6. General grammatical errors: Extra periods, golden should be just gold.

Response: In the revised manuscript we used an AI program (Grammarly) to suggest corrections and adjusted the text. We changed golden to gold.

7. If 3 samples were accidently pretreated with etoh, why did the authors just not use them in this study? 

Response: We have used them in the study, they are samples 8, 9, and 10.

It would have been better if 10 were split and then as a second arm of the study, select another 10 specimens, preincubate with etoh and then split into nfpe and ffpe. This would have made the study and data stronger. 

Response: We agree that it would have been better to have more samples. This was set up as a pilot study and we only used a limited amount of tissues. We did not know there would be a large effect of 35% EtOH 5 (now 6) years ago. 

It should be addressed why this decision to keep these specimens was made. (I also hope that the lab situation where issues like accidently putting tissues into wrong containers has been rectified as to not affect patient care.) 

Response: The 3 tissues were put in EtOH by an intern until the responsible pathologist returned from a seminar. We did not use the EtOH-treated samples for diagnostics. We now mention this in the manuscript. We did not throw them away because we only had a limited amount of samples available for the study.

---

## [Decision Letter · Decision Letter 1]

20 Nov 2024

PONE-D-24-30519R1Formalin-free tissue embedding is less hazardous and results in better DNA qualityPLOS ONE

Dear Dr. Hoogland,

Thank you for submitting your manuscript to PLOS ONE. After careful consideration, we feel that it has merit but does not fully meet PLOS ONE’s publication criteria as it currently stands. Therefore, we invite you to submit a revised version of the manuscript that addresses the points raised during the review process.

We look forward to receiving your revised manuscript.

Kind regards,

Sherin Reda Rouby, PhD

Academic Editor

PLOS ONE

Journal Requirements:

Reviewers' comments:

Reviewer's Responses to Questions

Comments to the Author

1. If the authors have adequately addressed your comments raised in a previous round of review and you feel that this manuscript is now acceptable for publication, you may indicate that here to bypass the “Comments to the Author” section, enter your conflict of interest statement in the “Confidential to Editor” section, and submit your "Accept" recommendation.

Reviewer #1: All comments have been addressed

Reviewer #2: All comments have been addressed

2. Is the manuscript technically sound, and do the data support the conclusions?

Reviewer #1: Yes

Reviewer #2: Yes

3. Has the statistical analysis been performed appropriately and rigorously? 

Reviewer #1: Yes

Reviewer #2: Yes

4. Have the authors made all data underlying the findings in their manuscript fully available?

Reviewer #1: Yes

Reviewer #2: Yes

5. Is the manuscript presented in an intelligible fashion and written in standard English?

Reviewer #1: Yes

Reviewer #2: Yes

6. Review Comments to the Author

Reviewer #1: 1. Tables: The letter (S) written beside patient number 4 stands for what?

2. Table 1, uterus sample 10, there is a typo error in the ratio column (58 instead of 5,8).

Reviewer #2: List of punctuation corrected:

-supercritical carbon dioxide (CO2), basically,

-In our study, fresh, unfixed tissues arriving from the operation room

-2 x Paraffin

-new paragraph starting with line - For the NFPE route, fresh, unfixed tissue was loaded in the vessels

-the (tissue) vessels

-new paragraph starting with line - For both FFPE and NFPE, after embedding at the

-Wong et al.6.

-discounts, etcetera

-(The Netherlands)

-Results section 1st paragraph: no need to spell out abbreviations again

-Move figure 1 to methods section

-"bad block"

-been reported in some studies6,14,15, whereas others report hardly any artifacts

-and 400bp (AF4), respectively, as described by

-This sentence is confusing and needs to be re-worded to make it clear - DNA was isolated from the 10 slides between the before and after HE and the DNA concentration was measured.

7. PLOS authors have the option to publish the peer review history of their article (what does this mean?). If published, this will include your full peer review and any attached files.

Do you want your identity to be public for this peer review? For information about this choice, including consent withdrawal, please see our Privacy Policy.

Reviewer #1: Yes: Prof0. Dr. Amal Abd E l-hafez

Reviewer #2: No

---

## [Author Response · Author response to Decision Letter 1]

2 Dec 2024

Reviewers' comments:

Reviewer's Responses to Questions

Comments to the Author

1. If the authors have adequately addressed your comments raised in a previous round of review and you feel that this manuscript is now acceptable for publication, you may indicate that here to bypass the “Comments to the Author” section, enter your conflict of interest statement in the “Confidential to Editor” section, and submit your "Accept" recommendation.

Reviewer #1: All comments have been addressed

Reviewer #2: All comments have been addressed

2. Is the manuscript technically sound, and do the data support the conclusions?

Reviewer #1: Yes

Reviewer #2: Yes

3. Has the statistical analysis been performed appropriately and rigorously? 

Reviewer #1: Yes

Reviewer #2: Yes

4. Have the authors made all data underlying the findings in their manuscript fully available?

Reviewer #1: Yes

Reviewer #2: Yes

5. Is the manuscript presented in an intelligible fashion and written in standard English?

Reviewer #1: Yes

Reviewer #2: Yes

6. Review Comments to the Author

Reviewer #1: 1. Tables: The letter (S) written beside patient number 4 stands for what? 

Patient 4 was a deceased (S) patient where 4 tissues were harvested. We now mention that in the table 1. text.

2. Table 1, uterus sample 10, there is a typo error in the ratio column (58 instead of 5,8).

There is no typo, DNA concentration was 33,38 times higher wheareas the cell surface ratio was 0,572. SO 33,38/0,572=58. In this sample the qPCR ratio was also exceptionally high (table 2). 

Reviewer #2: List of punctuation corrected:

-supercritical carbon dioxide (CO2), basically,

We corrected this according to the suggestion of reviewer 2

-In our study, fresh, unfixed tissues arriving from the operation room

We corrected this according to the suggestion of reviewer 2

-2 x Paraffin

We corrected this according to the suggestion of reviewer 2

-new paragraph starting with line - For the NFPE route, fresh, unfixed tissue was loaded in the vessels

We corrected this according to the suggestion of reviewer 2

-the (tissue) vessels

We corrected this according to the suggestion of reviewer 2

-new paragraph starting with line - For both FFPE and NFPE, after embedding at the

We corrected this according to the suggestion of reviewer 2

-Wong et al.6.

We corrected this according to the suggestion of reviewer 2

-discounts, etcetera

We corrected this according to the suggestion of reviewer 2

-(The Netherlands)

We corrected this according to the suggestion of reviewer 2

-Results section 1st paragraph: no need to spell out abbreviations again

As suggested by the reviewer, we do not spell out abbreviations in paragraph 1 of the results section.

-Move figure 1 to methods section

We moved figure 1 to the methods section

-"bad block"

We corrected this according to the suggestion of reviewer 2

-been reported in some studies6,14,15, whereas others report hardly any artifacts

We corrected this according to the suggestion of reviewer 2

-and 400bp (AF4), respectively, as described by

We corrected this according to the suggestion of reviewer 2

-This sentence is confusing and needs to be re-worded to make it clear - DNA was isolated from the 10 slides between the before and after HE and the DNA concentration was measured.

We re-worded the paragraph to make it more clear

7. PLOS authors have the option to publish the peer review history of their article (what does this mean?). If published, this will include your full peer review and any attached files.

Do you want your identity to be public for this peer review? For information about this choice, including consent withdrawal, please see our Privacy Policy.

Reviewer #1: Yes: Prof0. Dr. Amal Abd E l-hafez

Reviewer #2: No

---

## [Editor Report · Decision Letter 2]

6 Dec 2024

Formalin-free tissue embedding is less hazardous and results in better DNA quality

PONE-D-24-30519R2

Dear Dr. Hoogland,

We’re pleased to inform you that your manuscript has been judged scientifically suitable for publication and will be formally accepted for publication once it meets all outstanding technical requirements.

Kind regards,

Sherin Reda Rouby, PhD

Academic Editor

PLOS ONE
---

## [Editor Report · Acceptance letter]

11 Dec 2024

PONE-D-24-30519R2 

PLOS ONE

Dear Dr. Hoogland, 

I'm pleased to inform you that your manuscript has been deemed suitable for publication in PLOS ONE. Congratulations! Your manuscript is now being handed over to our production team.

Kind regards, 

on behalf of

Professor Sherin Reda Rouby 

Academic Editor

PLOS ONE